# Statistical Learning and Inverse Problems: A Stochastic Gradient Approach

**Yuri R. Fonseca**
Decision, Risk and Operations
Columbia University
New York, NY
yfonseca23@gsb.columbia.edu

**Yuri F. Saporito**
School of Applied Mathematics
Getulio Vargas Foundation
Rio de Janeiro, Brazil
yuri.saporito@fgv.br

## Abstract

Inverse problems are paramount in Science and Engineering. In this paper, we consider the setup of Statistical Inverse Problem (SIP) and demonstrate how Stochastic Gradient Descent (SGD) algorithms can be used to solve linear SIP. We provide consistency and finite sample bounds for the excess risk. We also propose a modification for the SGD algorithm where we leverage machine learning methods to smooth the stochastic gradients and improve empirical performance. We exemplify the algorithm in a setting of great interest nowadays: the Functional Linear Regression model. In this case we consider a synthetic data example and a classification problem for predicting the main activity of bitcoin addresses based on their balances.

## 1 Introduction

Inverse Problems (IP) might be described as the search of an unknown parameter (that could be a function) that satisfies a given, known equation. Considering the notation:

$$y = A[f^\circ] + \text{ noise},$$

where $f^\circ$ and $y$ are elements of given Hilbert spaces, we would like to compute (or estimate) $f^\circ$ given the data $y$ for some level of noise. Typically, IPs are ill-posed in the sense that the solution does not depend continuously on the data. There are several very important and impressive examples of IPs in our daily lives. Medical imaging has been using IPs for decades and it has shaped the area, as for instance, Computerized Tomography (CT) and Magnetic Resonance Imaging (MRI). For an introductory text, see Vogel [2002].

A vast literature of IPs is devoted to deterministic problems where the noise term is also a element of a Hilbert space and commonly assumed small in norm, which is not usually verifiable in practice. In this work, we will take a different avenue, known as Statistical Inverse Problems (SIP). This approach is a formalization of IPs within a probabilistic setting, where the uncertainty of all measurements are properly considered. Our focus in this work is to propose a direct and practical method for solving SIP problems and, at the same time, provide theoretical guarantees for the excess risk performance of the algorithm we develop. Our algorithm is based on a gradient descent framework, where stochastic gradients (or base learners that approximate the stochastic gradients) are used to estimate general functional parameters.

The paper is organized as follows. We finish this section contextualizing our paper in the broad literature and stating our main contributions. In Section 2, we formally introduce the learning problem that we analyze. In Section 3, we provide examples of practical problems that fits within our formulation. In Section 4 we provide our main results and algorithms. Finally, in Section 5 we provide numerical examples and a real data application for a Functional Linear Regression problem

36th Conference on Neural Information Processing Systems (NeurIPS 2022).

(FLR). Due to space constraints, some of the figures, proofs and experiments were moved to the supplementary material.

## 1.1 Contribution

We provide a novel numerical method to estimate functional parameters in SIP problems using stochastic gradients. More precisely, we extend the properties and flexibility of SGD and boosting algorithms to a broader class of problems by bridging the gap between the IP and machine learning communities.

Whereas most of the IPs methods focus on regularization strategies to "invert" the operator $A$, we propose a gradient descent type of algorithm to estimate the functional parameter directly. Our algorithm works in the same spirit as Stochastic Gradient Descent algorithms with sample averaging. While results of SGD are well understood in the context of regression problems in finite and infinite dimensions and SGD is well understood in deterministic IPs, SGD have not yet been considered under the SIP formulation.

We show that our procedure also ensures risk consistency in expectation and high probability under the statistical setting. Furthermore, we propose a modification in our algorithm to substitute the stochastic gradients by base learners similarly to boosting algorithms, Mason et al. [1999], Friedman [2001]. This modification improve a common challenge faced by SIP problems: the discretization procedures of the operator $A$ that arises in SIP.

## 1.2 Literature Review

Historically, SIP was first introduced in Sudakovand and Khalfin [1964] where IPs from Mathematical Physics were recast into a statistical framework. For a more structured introduction, we forward the reader to Kaipio and Somersalo [2004]. Several advances were made in the parametric approach to SIP, where the unknown function is assumed to be completely described by an unknown parameter living in a finite dimensional space, see for instance Evans and Stark [2002]. In our paper, however, we will consider the nonparametric framework as described in Cavalier [2008]. In this setting, we see the IP as a search of an element of an infinite dimensional space.

When considering IPs (and SIP, in particular), there are several ways to regularize the problem in order to deal with its ill-posedness. For instance, one could consider roughness penalty or a functional basis as in Tenorio [2001]. Additionally, one could examine Tikhonov and spectral cut-off regularizations as in Bissantz et al. [2007]. For many of those standard approaches, consistency under the SIP setting and rates of convergences were established. See for instance Bissantz et al. [2004], Bissantz and Holzmann [2008]. A thoroughly discussion of stochastic gradient algorithms is outside the scope of this work and we refer the reader to Zinkevich [2003], Nesterov et al. [2018] and references therein.

There have been numerous applications of Machine Learning (and Deep Learning, in particular) to solve IPs, in recent years. However, in our opinion, these applications are akin of the capabilities that these new techniques could bring to this area of research. Some attention have been given to imaging problems as in Jin et al. [2017] and Ongie et al. [2020]. Under deterministic IP, the paper Li et al. [2020] studies the regularization and convergence rates of penalized neural networks when solving regression problems. See also Adler and Öktem [2017] and Bai et al. [2020]. Other important references regarding SGD for deterministic IP are Jin et al. [2021], Tang et al. [2019], Jin et al. [2020].

The main examples we bring in our paper is the class of Functional Linear Regression (FLR). This problem has drawn the attention of the statistical, econometric and computer science communities in the past decade, see Cai and Hall [2006], Yao et al. [2005], Hall and Horowitz [2007]. The usual methodology applied to this problem is the well-known FDA. For example, one could consider a prespecified functional basis to regularize the regression problem Goldsmith et al. [2011] or one could use the Functional Principal Component (FPC) basis, Morris [2015]. More recently, methods inspired in machine learning for standard linear regression problems were also extended to the FLR setting, see for instance James et al. [2009], Fan et al. [2015] for methods that are suitable for high dimensional covariates or interpretable in the LASSO sense. In this work we show how our modification to the SGD algorithm can be seen as an averaging of boosting estimators and can also be used to estimate FLR models in the high-dimensional setting.

## 2 Problem Formulation

We start by fixing a probability space $(\Omega, \mathcal{A}, \mathbb{P})$ and a vector space $\mathbb{X}$ of inputs. We denote the random input by $\mathbf{X} \in L^2(\Omega, \mathcal{A}, \mathbb{P})$ taking values in $\mathbb{X}$ and consider the space [1]$L^2(\mathbb{X}, \mathcal{B}(\mathbb{X}), \mu_{\mathbf{X}})$, henceforth referred to as $L^2(\mathbb{X})$, of functions $g : \mathbb{X} \longrightarrow \mathbb{R}^d$ with inner product $\langle g_1, g_2 \rangle_{L^2(\mathbb{X})} = \mathbb{E}[\langle g_1(\mathbf{X}), g_2(\mathbf{X}) \rangle]$ and norm $\|g\|^2_{L^2(\mathbb{X})} = \mathbb{E}[\|g(\mathbf{X})\|^2] < +\infty$, where $\langle \cdot, \cdot \rangle$ and $\| \cdot \|$ are the inner product and norm of $\mathbb{R}^d$.

We also consider a Hilbert space $\mathbb{H}$ with inner product $\langle \cdot, \cdot \rangle_{\mathbb{H}}$. Finally, we consider an operator $A : \mathbb{H} \longrightarrow L^2(\mathbb{X})$. This operator defines a direct problem and we assume that it is known. Given $f \in \mathbb{H}$, we use the notation $A[f] \in L^2(\mathbb{X})$, i.e. $A[f]$ is a square-integrable function $A[f] : \mathbb{X} \longrightarrow \mathbb{R}^d$.

We are interested in solving the statistical inverse problem related to $A$: jointly to observing samples of $\mathbf{X}$ taking values in $\mathbb{X}$, we observe noisy samples of $A[f^\circ](\mathbf{X})$, for some fixed, unknown $f^\circ \in \mathbb{H}$, which we denote by $\mathbf{Y}$:

$$\mathbf{Y} = A[f^\circ](\mathbf{X}) + \epsilon, \tag{1}$$

where $\epsilon$ is a zero-mean random noise. The problem we will pore over in this paper is the estimation of $f^\circ$ based on this given sample.

Let $\ell : \mathbb{R}^d \times \mathbb{R}^d \to \mathbb{R}_+$ be a point-to-point loss function as, for example, the squared loss $\ell(\mathbf{y}, \mathbf{y}') = \frac{1}{2}\|\mathbf{y} - \mathbf{y}'\|^2$, for regression, or the logistic loss function $\ell(\mathbf{y}, \mathbf{y}') = \log(1 + e^{-\mathbf{y} \cdot \mathbf{y}'})$, for classification. We define the populational risk as:

$$\mathcal{R}_A(f) \triangleq \mathbb{E}[\ell(\mathbf{Y}, A[f](\mathbf{X}))],$$

and we would like to solve:

$$\inf_{f \in \mathcal{F}} \mathcal{R}_A(f), \tag{2}$$

where $\mathcal{F} \subset \mathbb{H}$ with $f^\circ \in \mathcal{F}$. We will denote by $\partial_2$ the partial derivative with respect to the second argument.

Given a sample, we will study how to control the excess risk of a functional estimator $\hat{f}$ of $f^\circ$:

$$\mathcal{R}_A(\hat{f}) - \inf_{f \in \mathcal{F}} \mathcal{R}_A(f). \tag{3}$$

Instead of taking the standard route of solving the Empirical Risk Minimization problem and later establishing results for (3), in Section 4 we show how our algorithms allow us to tackle (3) directly by constructing stochastic gradients directly for the populational risk.

## 3 Examples: motivation

Before we formalize our results, we first motivate the study of Eq. (1) with a few of applications. Each of those problems have a myriad of solutions on their own. For more information on those IPs, see, for instance, Vogel [2002].

**Deconvolution**. This type of inverse problems relate the values of $\mathbf{Y}$ and $\mathbf{X}$ through the following convolution equation:

$$\mathbf{Y} = \int_{\mathbb{W}} k(\mathbf{X} - \mathbf{w}) f(\mathbf{w}) d\mu(\mathbf{w}) + \epsilon,$$

where $\mathbb{X} = \mathbb{W} = \mathbb{R}^d$, $\mathbb{H} = L^2(\mathbb{W}, \mathcal{B}, \mu)$ and the kernel $k : \mathbb{R}^d \longrightarrow \mathbb{R}$ is known. In this case, we define the operator $A$ as:

$$A[f](\mathbf{x}) = \int_{\mathbb{W}} k(\mathbf{x} - \mathbf{w}) f(\mathbf{w}) d\mu(\mathbf{w}).$$

**Functional Linear Regression**. Consider the scalar, multivariate functional linear regression: let $\mathbb{X} = D([0, T])$ (the space of right-continuous with left limits functions taking values in $\mathbb{R}^{d \times k}$ with

---

[1]We denote by $\mathcal{B}(\mathbb{X})$ the Borel sigma algebra in $\mathbb{X}$ and by $\mu_{\mathbf{X}}$ the distribution of the random variable $\mathbf{X}$.

the sup norm) so that $L^2(\Omega)$ is the space of stochastic processes with sample paths in $D([0,T])$ and norm

$$\|\mathbf{X}\|^2 = \mathbb{E}\left[\sup_{t\in[0,T]} \|\mathbf{X}(t)\|^2\right]. \tag{4}$$

Moreover, $\mathbb{H} = L^2([0,T])$ taking values in $\mathbb{R}^k$ and $\mathbf{Y}$ is given by the following model

$$\mathbf{Y} = \int_0^T \mathbf{X}(s)f(s)ds + \epsilon,$$

where $f \in \mathbb{H}$. Here we changed the notation from $\mathbf{w}$ to $s$ in order to keep the classical notation from FLR. In this case,

$$A[f](\mathbf{x}) = \int_0^T \mathbf{x}(s)f(s)ds.$$

The model can be easily extended to deal with $\mathbf{Y}$ taking label values such as in a classification problem as we will show in the numerical studies.

Due to space constraints, we provide examples in the FLR setting. In the supplementary material, we demonstrate how the algorithms presented in Section 4.3 can also be applied in deconvolution problems.

## 4  Theoretical Results and Algorithms

In this paper, we consider the following set of assumptions.

**Assumption 4.1.**

1. $A : \mathbb{H} \longrightarrow L^2(\mathbb{X})$ is a linear, bounded operator;

2. $\ell$ is a convex and $C^2$ function in its second argument;

3. There exists $\theta_0 > 0$ such that, for all $f, g \in \mathbb{H}$,

$$\sup_{|\theta|\leq\theta_0} \mathbb{E}\left[\left\langle A[g](\mathbf{X}),\ \partial_{22}\ell\big(Y, A[f](\mathbf{X}) + \theta A[g](\mathbf{X})\big)A[g](\mathbf{X})\right\rangle\right] < \infty;$$

4. $f^\circ \in \arg\min_{f\in\mathcal{F}} \mathcal{R}_A(f)$ and $\mathcal{R}_A(f^\circ) > -\infty$;

5. $\sup_{f,f'\in\mathcal{F}} \|f - f'\|_{L^2(\mathbb{W})} = D < \infty$.

Assumption 1 is our strongest one, since it imposes that our operator is linear and bounded. Nevertheless, linear SIPs encompass a wide class of problems of practical and theoretical interest for engineering, statistics and computer science communities among others, a few of them presented in Section 3. Moreover, the nonlinear case could be similarly studied with more cumbersome notation and assumptions. Assumption 2 is standard for gradient based algorithms and is commonly assumed in many learning problems. Assumption 3 is a mild integrability condition of the loss function commonly satisfied in many practical situations. For instance, in the squared loss case, this assumption becomes $\mathbb{E}[\|A[g](\mathbf{X})\|^2] < \infty$, which is automatically satisfied since $A[g] \in L^2(\mathbb{X})$. Assumption 4 is needed so the problem we analyze indeed has a solution. Assumption 5 is stating that the diameter of the set $\mathcal{F}$ is finite.

One should notice that our set of assumptions does include the class of ill-posed (linear) inverse problems since we do not need to assume that $A$ is bijective. If that were the case, it is known that then $A$ would have a bounded inverse, and then, the IP would not be ill-posed.

In the next sections we provide our theoretical results. Instead of following the common approach of minimizing the Empirical Risk Minimization problem, we show how to compute stochastic gradients in order to control directly for the excess risk (3) both in expectation and in probability.

## 4.1 Preliminaries

Our first result allows us to compute the gradient of the populational risk at a given functional parameter $f$. Before we present it, note that, by linearity, $A : \mathbb{H} \to L^2(\mathbb{X})$ is differentiable and, for every $f, g \in \mathbb{H}$, we have that the directional derivative of $A[f]$ in the direction $g$ is given by

$$DA[f](g) = \lim_{\delta \to 0} \frac{1}{\delta}\left(A[f + \delta g] - A[f]\right) = A[g].$$

Note that the directional derivative does not depend on the point $f$ that we are evaluating the gradient. Let $A^*$ denote the adjoint operator of $A$ defined as the linear and bounded operator $A^* : L^2(\mathbb{X}) \longrightarrow \mathbb{H}$ such that[2]

$$\langle A[f], h\rangle_{L^2(\mathbb{X})} = \langle f, A^*[h]\rangle_{\mathbb{H}}, \text{ for all } f \in \mathbb{H} \text{ and } h \in L^2(\mathbb{X}).$$

The following lemma holds true:

**Lemma 4.2.** *Under 1, 2 and 3 of Assumption 4.1 we have that*

$$\nabla \mathcal{R}_A(f) = A^*[\phi_f] \in \mathbb{H},$$

*where $\phi_f(\mathbf{x}) = \mathbb{E}[\partial_2 \ell(\mathbf{Y}, A[f](\mathbf{x})) \mid \mathbf{X} = \mathbf{x}]$.*

*Proof.* Firstly, define, for fixed $(\mathbf{X}, \mathbf{Y})$,

$$\psi(\delta) = \ell(\mathbf{Y}, A[f + \delta g](\mathbf{X})) = \ell(\mathbf{Y}, A[f](\mathbf{X}) + \delta A[g](\mathbf{X})).$$

Then, we get the directional derivative of the risk function in direction $g$ by applying the Taylor formula for $\psi$ as a function of $\delta$ around $\delta = 0$:

$$
\begin{aligned}
D\mathcal{R}_A(f)(g) &= \lim_{\delta \to 0} \frac{1}{\delta}\left(\mathcal{R}_A(f + \delta g) - \mathcal{R}_A(f)\right) \\
&= \lim_{\delta \to 0} \mathbb{E}\left[\frac{1}{\delta}\left(\delta\Big\langle \partial_2\ell(\mathbf{Y}, A[f](\mathbf{X})), A[g](\mathbf{X})\Big\rangle \right.\right. \\
&\qquad\qquad \left.\left. + \tfrac{1}{2}\delta^2 \Big\langle A[g](\mathbf{X}), \, \partial_{22}\ell(\mathbf{Y}, A[f](\mathbf{X}) + \theta A[g](\mathbf{X})), A[g](\mathbf{X})\Big\rangle\right)\right],
\end{aligned}
$$

where $\theta$ comes from the Taylor formula and it is between $-\theta_0$ and $\theta_0$, for some fixed $\theta_0 > 0$. Hence, by Assumption 3, we find

$$D\mathcal{R}_A(f)(g) = \mathbb{E}\left[\Big\langle \partial_2\ell(\mathbf{Y}, A[f](\mathbf{X})), A[g](\mathbf{X})\Big\rangle\right].$$

By the definition of $\phi$ and by conditioning in $\mathbf{X}$, we find

$$D\mathcal{R}_A(f)(g) = \mathbb{E}[\langle \phi_f(\mathbf{X}), A[g](\mathbf{X})\rangle] = \langle \phi_f, A[g]\rangle_{L^2(\mathbb{X})} = \langle A^*[\phi_f], g\rangle_{\mathbb{H}}.$$

Finally, we get that the descent direction $\nabla\mathcal{R}_A(f)$ is given by $A^*[\phi_f] \in \mathbb{H}$. $\qquad\square$

## 4.2 Unbiased Estimator of the Gradient

In order to define an unbiased estimator of the gradient $\nabla\mathcal{R}_A$, we consider the following assumption:

**Assumption 4.3.**

1. $\mathbb{H}$ is a Hilbert space of functions from $\mathbb{W}$ to $\mathbb{R}^k$;

2. There exists a kernel $\Phi : \mathbb{X} \times \mathbb{W} \longrightarrow \mathbb{R}^{k \times d}$ such that

$$A^*[h](\mathbf{w}) = \mathbb{E}[\Phi(\mathbf{X}, \mathbf{w})h(\mathbf{X})] \in \mathbb{H},$$

for all $\mathbf{w} \in \mathbb{W}$ and $h \in L^2(\mathbb{X})$.

Several examples, including the Functional Linear Regression, as we will verify in Section 5, satisfy this assumption. Additionally, there are two situations that encompass many important SIPs. The first one is a restriction of the Hilbert space $\mathbb{H}$ without restrictions on the operator $A$:

---

[2]The adjoint of a linear, bounded operator always exists.

**Lemma 4.4.** *Assumption 4.3 is verified if $\mathbb{H}$ is a Reproducing Kernel Hilbert Space (RKHS).*

*Proof.* For simplicity of notation, we assume $k = 1$. Notice that, by the RKHS assumption, $\varphi_{\mathbf{w}} : L^2(\mathbb{X}) \longrightarrow \mathbb{R}$ defined as $\varphi_{\mathbf{w}}(h) = A^*[h](\mathbf{w})$, for $h \in L^2(\mathbb{X})$, is an element of the dual of $L^2(\mathbb{X})$, i.e. there existis $M_{\mathbf{w}} < +\infty$ such that

$$|\varphi_{\mathbf{w}}(h)| = |A^*[h](\mathbf{w})| \leq M_{\mathbf{w}} \|A^*[h]\|_{\mathbb{H}} \leq M_{\mathbf{w}} \|A^*\| \|h\|_{L^2(\mathbb{X})}.$$

Hence, by the Riesz Representation Theorem, there exists a kernel $\Phi(\cdot; \mathbf{w}) : \mathbb{X} \longrightarrow \mathbb{R}^d$ such that, for all $h \in L^2(\mathbb{X})$,

$$A^*[h](\mathbf{w}) = \varphi_{\mathbf{w}}(h) = \langle \Phi(\cdot; \mathbf{w}), h \rangle_{L^2(\mathbb{X})} = \mathbb{E}[\langle \Phi(\mathbf{X}, \mathbf{w}), h(\mathbf{X}) \rangle],$$

as desired. $\square$

*Remark* 4.5. If the RKHS $\mathbb{H}$ has kernel $K$, then $\Phi$ is given by $\Phi(\mathbf{x}, \mathbf{w}) = A[K(\cdot, \mathbf{w})](\mathbf{x})$. Indeed, by the definition of kernel in the RKHS and the definition of $A^*$, we find

$$A^*[h](\mathbf{w}) = \langle A^*[h], K(\cdot, \mathbf{w}) \rangle_{\mathbb{H}} = \langle h, A[K(\cdot, \mathbf{w})] \rangle_{L^2(\mathbb{X})}.$$

The second situation considers a different (and somewhat less restrictive) assumption for the Hilbert space $\mathbb{H}$ and a particular, yet very general, class of operators $A$, called integral operators.

**Lemma 4.6.** *If $\mathbb{H} = L^2(\mathbb{W}, \mathcal{B}, \mu)$ taking values in $\mathbb{R}^k$ and $A$ is a integral operator of the form:*

$$A[f](\mathbf{x}) = \int_{\mathbb{W}} \varphi(\mathbf{x}, \mathbf{w}) f(\mathbf{w}) d\mu(\mathbf{w}),$$

*where $\varphi$ is a kernel taking values in $\mathbb{R}^{d \times k}$ such that $\varphi(\mathbf{x}, \cdot) f \in L^1(\mathbb{W}, \mathcal{B}, \mu)$, then Assumption 4.3 is verified.*

*Proof.* By the definition of the adjoint operator, we find

$$\langle f, A^*[h] \rangle_{L^2(\mathbb{W})} = \langle A[f], h \rangle_{L^2(\mathbb{X})} = \mathbb{E}[\langle A[f](\mathbf{X}), h(\mathbf{X}) \rangle]$$

$$= \mathbb{E}\left[\left\langle \int_{\mathbb{W}} \varphi(\mathbf{X}, \mathbf{w}) f(\mathbf{w}) d\mu(\mathbf{w}), h(\mathbf{X}) \right\rangle\right]$$

$$= \int_{\mathbb{W}} \mathbb{E}\left[\langle \varphi(\mathbf{X}, \mathbf{w}) f(\mathbf{w}), h(\mathbf{X}) \rangle\right] d\mu(\mathbf{w}) = \langle f, \mathbb{E}\left[\varphi(\mathbf{X}, \cdot)^T h(\mathbf{X})\right] \rangle_{L^2(\mathbb{W})}.$$

Hence, we conclude $A^*[h](\mathbf{w}) = \mathbb{E}\left[\varphi(\mathbf{X}, \mathbf{w})^T h(\mathbf{X})\right]$, which implies Assumption 4.3 with kernel $\Phi(\mathbf{x}, \mathbf{w}) = \varphi(\mathbf{x}, \mathbf{w})^T$. $\square$

*Remark* 4.7. The class of integral operators delivers several of the most important linear IPs. Additionally, using Green's function formulation, some PDEs IPs could also be recast as integral equations. For instance, the recovery the initial condition of a linear PDE with known Green function and observing the solution of the PDE at some future, fixed time.

Under Assumption 4.3, Lemma 4.2 implies the following very useful result that is the cornerstone of our method.

**Corollary 4.8.** *If Assumption 4.3 is verified, then the gradient of the risk function with respect to $f$ is given by*

$$\nabla \mathcal{R}_A(f)(\mathbf{w}) = \mathbb{E}[\Phi(\mathbf{X}, \mathbf{w}) \phi_f(\mathbf{X})].$$

Because of the results above, it is possible to construct an unbiased estimator for the gradient for the risk function for any $f$. In fact, for a given sample $(\mathbf{x}, \mathbf{y})$ and a fixed function $f$, we define, for any $\mathbf{w} \in \mathbb{W}$,

$$u_f(\mathbf{w}; \mathbf{x}, \mathbf{y}) = \Phi(\mathbf{x}, \mathbf{w}) \partial_2 \ell(\mathbf{y}, A[f](\mathbf{x})). \tag{5}$$

Therefore, conditioning in $\mathbf{X}$, we find

$$\mathbb{E}[u_f(\mathbf{w}; \mathbf{X}, \mathbf{Y})] = \mathbb{E}[\Phi(\mathbf{X}, \mathbf{w}) \partial_2 \ell(\mathbf{Y}, A[f](\mathbf{X}))] = \mathbb{E}[\mathbb{E}[\Phi(\mathbf{X}, \mathbf{w}) \partial_2 \ell(\mathbf{Y}, A[f](\mathbf{X})) \mid \mathbf{X}]]$$

$$= \mathbb{E}[\Phi(\mathbf{X}, \mathbf{w}) \mathbb{E}[\partial_2 \ell(\mathbf{Y}, A[f](\mathbf{X})) \mid \mathbf{X}]] = \mathbb{E}[\Phi(\mathbf{X}, \mathbf{w}) \phi_f(\mathbf{X})] = \nabla \mathcal{R}_A(f)(\mathbf{w}).$$

The main benefit is that with a single observation of $(\mathbf{X}, \mathbf{Y})$, we are able to compute an unbiased estimator for the gradient of the risk function under the true distribution.

## 4.3 Proposed Algorithms

Inspired by Corollary 4.8, we propose the following SGD algorithm for SIP problems that we called SGD-SIP: given an initial guess $f_0$, for each step $i$, we compute, following Eq. (5), an unbiased estimator $u_i$ for the gradient of the loss function. Next, we update an accumulated functional parameter by taking a stochastic gradient step in the direction of $u_i$ with step size $\alpha_i$. In the last step, we average all the accumulated gradient steps in the same spirit as Polyak and Juditsky [1992]. The choice of the step size needs to satisfy two criteria: $\sum_{i=1}^{n} \alpha_i$ sublinear in $n$, and $n\alpha_n \to \infty$ as $n \to +\infty$. We formally justify those desired properties in Theorem 4.9.

---

**Algorithm 1:** SGD-SIP

---

**input** : sample $\{\mathbf{x}_i, \mathbf{y}_i\}_{i=1}^{n}$, operator $A$, initial guess $f_0$

**output:** $\hat{f}_n$

$\hat{g}_0 = f_0$;

**for** $1 \leq i \leq n$ **do**

    Compute $u_i(\mathbf{w}) = \Phi(\mathbf{x}_i, \mathbf{w})\partial_2 \ell(\mathbf{y}_i, A[\hat{g}_{i-1}](\mathbf{x}_i))$;

    $\hat{g}_i = \hat{g}_{i-1} - \alpha_i u_i$;

**end**

Set $\hat{f}_n = \frac{1}{n} \sum_{i=1}^{n} \hat{g}_i$;

---

Algorithm 1 uses only one sample at a time in order to estimate the gradient of the true risk function. In order to preserve this property, we make the number of iterations equal to the sample size; it cannot be larger. This connects with the stopping rules in iterative algorithms in IP.

Algorithm 1 has a limitation common to many approaches to Inverse Problems: one cannot hope to compute $u_i$ for every possible $\mathbf{w}_i$ and some discretization of the operator $A$ is needed, see Kaipio and Somersalo [2007]. Since the SGD-SIP algorithm only computes the stochastic gradient in the points of discretization, it risks overfitting the data and provides non-smooth estimators. Next, we motivate Algorithm 2 in order to overcome the discretization problem by leveraging machine learning methods.

Consider that the space $\mathbb{W}$ was discretized in a grid of size $n_w$. In order to fully estimate the function $\hat{f}_n(\mathbf{w})$ for every $\mathbf{w} \in \mathbb{W}$, we consider a hypothesis class $\mathcal{H}$ and, in each step, we fit a function $\hat{h}_i^\star \in \mathcal{H}$ on the stochastic gradient $u_i$ in the discretized grid of $\mathbb{W}$. Note that in this case, $\mathcal{F}$ will be given by the linear span of the class $\mathcal{H}$. Each of these functions $h_i^\star$ can be seen as a base-learner in the same spirit of Boosting estimators, widely used in standard regression problem in the context of SIP Mason et al. [1999], Friedman [2001]. Next we present our algorithm ML-SGD.

---

**Algorithm 2:** ML-SGD

---

**input** : sample $\{\mathbf{x}_i, \mathbf{y}_i\}_{i=1}^{n}$, discretization $\{\mathbf{w}_j\}_{j=1}^{n_w}$ of $\mathbb{W}$, operator $A$, initial guess $f_0$

**output:** $\hat{f}_n$

$\hat{g}_0 = f_0$;

**for** $1 \leq i \leq n$ **do**

    **for** $1 \leq j \leq n_w$ **do**

        Compute $u_i(\mathbf{w}_j) = \Phi(\mathbf{x}_i, \mathbf{w}_j)\partial_2 \ell(\mathbf{y}_i, A[\hat{g}_{i-1}](\mathbf{x}_i))$;

    **end**

    $h_i^\star \in \arg\min_{h \in \mathcal{H}} \sum_{j=1}^{n_z} (u_i(\mathbf{w}_j) - h(\mathbf{w}_j))^2$;

    $\hat{g}_i = \hat{g}_{i-1} - \alpha_i h_i^\star$;

**end**

Set $\hat{f}_n = \frac{1}{n} \sum_{i=1}^{n} \hat{g}_i$;

---

The goal of ML-SGD is twofold. First, it allows us to interpolate the function $h_j^\star$ to points $\mathbf{w}$ not used in the discretization grid. Second, the ML procedure smooths the noise in each gradient step calculation leading to smoother approximations that helps avoiding over-fitting. We show in Section 5 and in the supplementary material the benefits of such an approximation when estimating the functional parameter $f^\circ$ in both simulated and empirical examples.

### 4.4 Main Result

Our main result is a finite sample bound for the expected excess risk of Algorithm 1. The result also extends to Algorithm 2 in the case where the base learner are also unbiased estimators.

**Theorem 4.9.** *Under Assumptions 4.1 and 4.3 and if the kernel $\Phi$ satisfies $C = \sup_{\mathbf{x} \in \mathbb{X}} \|\Phi(\mathbf{x}, \cdot)\|^2 < +\infty$[3], we have the following performance guarantee for Algorithm 1:*

$$\mathbb{E}\left[\mathcal{R}_A(\hat{f}_n) - \inf_{f \in \mathcal{F}} \mathcal{R}_A(f)\right] \leq \frac{D^2}{2n\alpha_n} + \frac{M(A, \mathcal{F})}{n} \sum_{i=1}^{n} \alpha_i,$$

*where $M(A, \mathcal{F}) = C(\mathbb{E}[\|\mathbf{Y}\|^2] + \|A\|^2 D^2) < \infty$.*

The proof of the theorem is provided in the supplementary material. Theorem 4.9 implies that if we pick the decreasing sequence $\{\alpha_i\}_{i=1}^n$ so that $n\alpha_n \to \infty$ ($\alpha_n$ cannot decrease too fast) but fast enough so that $\frac{1}{n} \sum_{i=1}^{n} \alpha_i \to 0$, then we get the convergence result. For instance, one could take $\alpha_i = \eta/\sqrt{i}$ for some fixed number $\eta$ normally taken to be in $(0, 1)$. In this case, the excess risk decreases in expectation with rate $O(1/\sqrt{n})$.

Theorem 4.9 also implies that the excess risk converges to zero in probability. For $\alpha_i = \eta/\sqrt{i}$ it is straightforward to check that

$$\limsup_{n \to +\infty} \mathbb{P}\left(\mathcal{R}_A(\hat{f}_n) - \inf_{f \in \mathcal{F}} \mathcal{R}_A(f) > 0\right) = 0.$$

Finite sample bounds with high probability can also be provided under stronger assumptions about the stochastic gradients. See for instance Nemirovski et al. [2009].

## 5 Functional Linear Regression: numerical studies

In this section, we provide two applications of the Functional Linear Regression problem. We first demonstrate the performance of both algorithms in simulated data and next we provide an example for generalized linear models, applied to an classification problem using bitcoin transaction data. In Appendix B and Appendix C we also provide an additional numerical study in a different type of Inverse Problem: The deconvolution problem.

As we have seen in Section 3, the operator in the FLR case is given by

$$A[f](\mathbf{x}) = \int_0^T \mathbf{x}(s)f(s)ds. \tag{6}$$

Remember that in this example we are denoting $\mathbf{w}$ by $s$. Hence, by Lemma 4.6, we find $A^*[g](s) = \mathbb{E}\left[\mathbf{X}^T(s) g(\mathbf{X})\right]$. One can easily verify that $A^*[g] \in \mathbb{H}$ by the assumption that the norm (4) of $\mathbf{X}$ is finite. Therefore, we have $\Phi(\mathbf{x}, s) = \mathbf{x}^T(s)$, and we find, as in Eq. (5),

$$u_i(s) = \mathbf{x}^T(s)\partial_2 \ell(\mathbf{y}_i, A[\hat{g}_{i-1}](\mathbf{x}_i)). \tag{7}$$

### 5.1 Synthetic Data

We will consider the simulation study presented in González-Manteiga and Martínez-Calvo [2011]. Specifically, we set $\mathbb{W} = [0, 1]$, $f^\circ(z) = \sin(4\pi z)$, and $\mathbf{X}$ simulated accordingly a Brownian motion in $[0, 1]$. We also consider a noise-signal ratio of 0.2. We generate 100 samples of $\mathbf{X}$ and $\mathbf{Y}$ with the integral defining the operator $A$ approximated by a finite sum of 1000 points in $[0, 1]$. We test for the same specification when $f^\circ(z)$ oscillates between 1 and $-1$ in the points $0.25, 0.5, 0.75, 1$. For the observed data used in the algorithm procedure, we consider a coarser grid where each functional sample is observed at only 100 equally-spaced times. For the ML-SGD algorithm, we used smoothing splines and regression trees as base learners. We compared the results with Penalized Functional Linear Regression (PFLR) with cubic splines and cross-validation to select the number of basis expansion; we also compare with Landweber iteration method. In order to fit the PFLR model, we

---

[3]This assumption is satisfied for all the examples analyzed in this paper.

used the package *refund* Goldsmith et al. [2021] available in R. Detailed numerical results with error bars are displayed in Appendix C showing that the ML-SGD algorithm is at least as competitive as a state-of-the-art tailored specifically to FLR problems and superior to Landweber iterations, a general method for Inverse Problems. In Figure 1, we can see that both SGD-SIP and Landweber iterations achieve similar fit performance. PFLR with cubic splines and degree of freedom of 20 achieved a similar performance than ML-SGD with splines with 10 degrees of freedom. Both essentially recovers the true function $c^\circ$ perfectly. Despite the smoothness of $f^\circ$, ML-SGD with regression trees with 30 terminal nodes was also able to approximate $f^\circ$ well. Here we focus only on the methods with best performance. ML-SGD with both regression trees and splines with 20 degrees of freedom were able to recover the true function as well as PFLR with cubic splines and 20 degrees of freedom. In the appendix, we provide bar plots with the MSE under both scenarios with error bars for different simulations. Both Landweber iterations and the SGD algorithm are noisier and seems to overfit the data. We refer the reader to Appendix C for a detailed comparison among the methods.

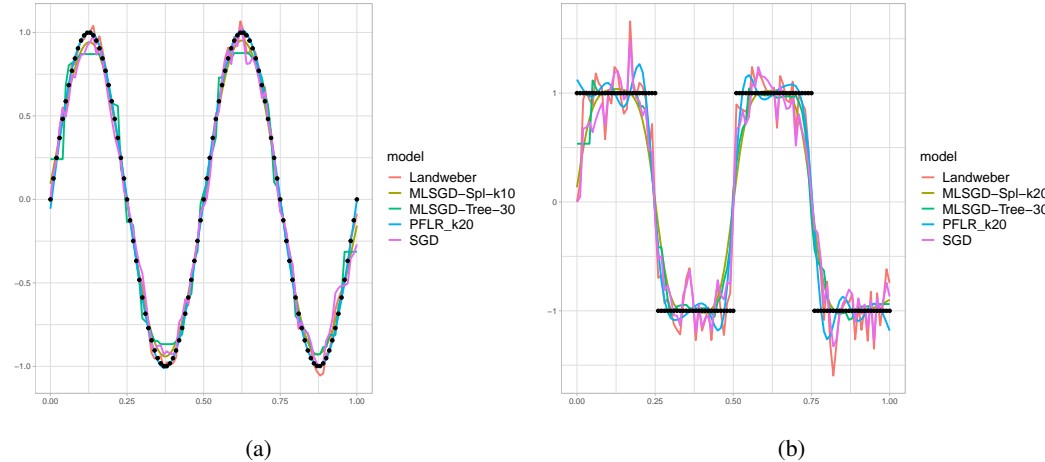

(a)                                                  (b)

Figure 1: Fitted results for synthetic data. True values for $f^\circ$ are displayed as black dots.

## 5.2    Real Data Application

Next we consider a classification problem in the FLR setting. The data set contains 3000 bitcoin addresses spanning from April 2011 and April 2017 and their respective cumulative credit, which is described as 501 equally spaced observations for the first 3000 hours of each address, normalized in the interval $[0, 1]$. For each address, we also have a label describing if the address was used for criminal activity, commonly called *darknet addresses*. In Table 1 we present a summary of the data. We refer the reader to Appendix A for more information about the data set used that we make available online.

Table 1: Summary information for the bitcoin wallet observations.

| category | obs | mean_initial_credit | mean_final_credit |
|---|---|---|---|
| Darknet Marketplace | 1494 | 242 | 1255 |
| Exchanges | 381 | 807 | 14278 |
| Gambling | 371 | 95 | 2289 |
| Pools | 366 | 1086 | 14071 |
| Services/others | 388 | 184 | 3883 |

Here we use the cumulative credit curve at each point in time as the explanatory variables $\mathbf{X} \in \mathbb{X} = D([0,1])$ and $Y \in \{-1, 1\}$ as the predicted outcome for the indicator variable that the category is *darknet* (addresses associated with illegal activities). We propose the following model: $\log \frac{P(Y=1|\mathbf{X})}{P(Y=-1|\mathbf{X})} = \int_0^T f(s)\mathbf{X}(s)ds$. By using the log-likelihood of the negative binomial, one can include its gradient with respect to the functional parameter directly in Equation (7) in order to use our framework. Other type of classification loss functions can also be used in the same spirit.

We compare the SGD-SIP and ML-SGD algorithm with PFLR. For the ML-SGD algorithm, we use two types of base learner, regression trees and cubic splines. The step sizes are taken to be equal of the form $O(1/\sqrt{i})$, where $i = 1, \cdots, n$ is the current step of the algorithm and $n$ is the total number of steps/sample. For the PFLR algorithm, we use cubic splines with different degrees of freedom and quadratic penalty term. We highlight that those choices of splines and penalty term are widely used in the literature, see, for instance, Goldsmith et al. [2011]. In Table 2 we provide 3-fold cross validation for the accuracy and kappa metrics. The SGD-SIP Algorithm achieved the best average performance in terms of accuracy. The same performance is achieved by the Functional PLR with cubic splines with number of knots equal to 20 and penalization for the derivative of the estimate. The ML-SGD algorithm with smooth splines with 20 degrees of freedom also achieved similar performance with a smoother estimator. Although the benchmark is as good as the ML-SGD algorithm, we highlight here that PFLR is tailored to Functional Data Analysis problem, while our approach is flexible for many different types of Linear SIP problems. Moreover, our algorithm can make use of only one sample at each iteration, which makes it suitable also for online applications. We refer the reader to the supplementary material for results under different choices of step size, number of knots and base functions for the PFLR model, other metrics and confusion matrices. In order to fit the PFLR mode, we used the package *refund* Goldsmith et al. [2021] available in R.

Table 2: Results for three fold cross-validation.

|  | fold_1 | fold_2 | fold_3 | avg_accuracy |
|---|---|---|---|---|
| ML-SGD-spline(k = 20) | 0.76 | 0.80 | 0.79 | 0.78 |
| ML-SGD-spline(k = 10) | 0.72 | 0.74 | 0.74 | 0.73 |
| ML-SGD-tree(depth = 20) | 0.78 | 0.77 | 0.78 | 0.78 |
| SGD-SIP | 0.79 | 0.77 | 0.82 | 0.79 |
| FPLR(k = 10) | 0.76 | 0.75 | 0.73 | 0.75 |
| FPLR(k = 20) | 0.77 | 0.81 | 0.80 | 0.79 |

## 6  Conclusion

In this work, we provided a novel numerical method to solve SIP based on stochastic gradients with theoretical guarantees for the excess risk. Moreover, we have shown how one can improve algorithmic performance by estimating base-learners for each stochastic gradient in the same spirit as boosting algorithms. Our framework can be applied in a variety of settings ranging from deconvolution problems, Functional Data analysis in both regression and classification problems, integral equations and other linear IPs related to PDEs. We demonstrate the performance of our method with numerical studies and also with a real world application data and comparing with widely used techniques in the FLR setting.

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
