# OpenReview forum: "Statistical Learning and Inverse Problems: A Stochastic Gradient Approach"
_NeurIPS.cc/2022/Conference — NeurIPS 2022 Accept_

### Official Review · Reviewer_Vx5e · 2022-06-29

**Rating:** 7
**Confidence:** 3
**Soundness:** 3 good
**Presentation:** 3 good
**Contribution:** 2 fair

**Summary:**

This paper mainly focuses on the formulation of SIP (statistical inverse problem) and the method to solve SIP.
In the first part of this paper, the authors formulate SIP utilizing functional analysis. They also give some examples of SIP under this formulation.

In the second part, after showing how to calculate the gradient under this setting, they propose two SGD-based algorithm to solve SIP. They give a finite sample bound for the expected excess risk of their algorithms.

In the last part, the authors provide numerical results for two applications of the Functional Linear Regression problem.

**Questions:**

1.In Lemma 4.3, you've mentioned that there exists a kernel \Phi. Can this kernel be easily found in many SIP problems (other than functional linear regression)?



**Limitations:**

It might be interesting if the results can be extended to nonlinear SIP (or any other relaxation of Assumption 4.1).

**Strengths And Weaknesses:**

Strengths: The authors give a novel numerical method to solve SIP, which shows good originality. They give both theoretical analysis and real data experiments, which shows the good quality of this paper.

Weakness: It seems that they do not show the intuition and advantage of their new method, which makes this paper somehow unclear. And the proof techniques are just classical convex optimization analysis. Therefore this paper lacks significance to some extent.

---

> ### Author Response · Authors · 2022-08-02
> **Referee 3**
>
> We are extremely grateful to the positive assessment and highly detailed and constructive feedback on our paper. In this rebuttal, we have addressed the comments of the review team. This has lead to multiple improvements including: Better exposition of the numerical studies, new applications, more general theoretical results and improved discussions about the framework we propose. Below we address each question and minor questions in a point by point fashion.
>
> Weakness:
>
> 1 - We believe that one of the main contribution is the flexible treatment for linear IPs in general. Many of these methods either needs tailored solutions for a particular problem, such as PFLR, or the general ones are ''known'' for their instability in applications, such as Landweber iterations. Here we provide a unified framework that generalizes from RKHS and allows one to use ''any'' type of ML method to solve a quite general class of SIP problems with statistical guarantees. Although the mathematical techniques used are not new, we believe that our contribution is to show that under statistical framework, the structure of IPs can be exploit in order to provide theoretical guarantees for our algorithms that directly holds for the populational risk, while previous analysis in the literature focus in understanding gradient descent approaches mostly for the empirical loss in the particular case of RKHS. Although the mathematical techniques used are not new, we believe that another aspect of our contribution is to show that under statistical framework, the structure of IPs can be exploit in order to provide theoretical guarantees for our algorithms that directly holds for the populational risk, while previous analysis in the literature focus in understanding gradient descent approaches mostly for the empirical loss in the particular case of RKHS.
>
> Questions:
>
> 1- Thank you for bringing this up. We have rewritten this part to make it clear how to find the kernel \Phi in several important examples (see Lemma 4.4 and 4.6).
>
> Limitations:
>
> 1- Lemma 4.2 could be generalized for the case of nonlinear Frechet differentiable A. In this case, the gradient of R_A is DA^*[phi_f]. Some interesting implications could be studied as future research, but since the convexity of R_A is lost, the analysis of this problem becomes much more difficult. Maybe a first step could be considering Monotone operators.

---

### Official Review · Reviewer_ea5U · 2022-07-11

**Rating:** 5
**Confidence:** 3
**Soundness:** 2 fair
**Presentation:** 2 fair
**Contribution:** 2 fair

**Summary:**

In this paper, the authors propose an SGD approach to solve statistical inverse problems (SIPs). Formally, the problem is to find the parameter $f$ satisfying the relation $Y = A\[f\](X) + \varepsilon$, where the operator $A$ is known, and $Y$ and $X$ are given observations. This can be achieved by minimizing the quantity $\mathcal{R}_A(f) = E\[ l(Y, A\[f \](X))\]$, where $l$ is a loss function. For $A$ linear and bounded, and with additional (milder) assumptions, the authors are able to write the gradient of $\mathcal{R}_A(f)$ as the average of a function $u_f$, whose expression has to be derived explicitly for the specific SIP considered. With this result at hand, the authors propose two algorithms to solve SIPs in the spirit of SGD, where the average giving the gradient is approximated by one sample at each iteration. Their main result proves a performance guarantee for such algorithms. The authors apply the proposed techniques to functional linear regressions, performing numerical tests both on synthetic and real data, and deconvolution (discussed in the supplementary material).

**Questions:**

I have the following questions/comments for the authors:

1. I would appreciate it if the authors could comment on my concerns in the previous section. I believe that addressing these points in the main text would be beneficial for the paper.

2. The presentation of numerical results could be improved. I believe that Figures 1 and 2 in section 5.2 are a little aimless, as they do not really support any claim in the paper (they take up quite some space, though). Moreover, the label font size in Figure 1 is too small and the x-axis has no scale (or ticks). It is unclear what's on the y-axis in Figure 2 ($f(s)$? If so, we don't know the real $f$ anyway...what is the point?). In any case, I would suggest combining the two figures and using the same format for both. The synthetic data experiments could be useful to test robustness against the signal-to-noise ratio.

3. The sentence 'We highlight that those choices of splines and penalty term are widely used in the literature', line 230, might need a reference.

4. I am not sure I understand what 'with only one observation' means in this sentence: 'The main benefit is that with only one observation we are able to compute an unbiased estimator for the gradient of the risk function under the true distribution', line 158/159. Can you clarify?

5. I would change the article 'An' in the title with 'A'.

6. Line 215 'where and each' -> 'where each'?

7. Line 103/104 'to control directly to tackle (3) directly' -> 'to tackle (3) directly'?

**Limitations:**

The authors have partially addressed the limitations of their work, though there is space for improvement (see the section Strengths And Weaknesses).

**Strengths And Weaknesses:**

The paper proposes an SGD approach to solve SIPs, thus providing, to the best of my knowledge, an original contribution (at least within the probabilistic formulation of inverse problems). The exposition is clear and compact (section 5 could be improved, though), and the related literature is well covered.

My main concerns with this work are the following:

1. General applicability of the proposed method. Even considering the (not so week) Assumption 4.1, the proposed method crucially depends on the possibility of identifying the kernel $\phi$ appearing in Lemma 4.3. The authors do not discuss the limitations associated with identifying $\phi$, and only present applications where this is doable. Moreover, the proposed algorithms are designed (or, at least, presented) following Corollary 4.4, i.e. considering the squared loss function. It is unclear to me if Theorem 4.5 applies only under this corollary.

2. Actual improvements over other methods. The comparisons proposed by the authors seem to suggest that their algorithms are at most as good as the FPLR (see Table 2). I wonder then what are the actual advantages of solving SIPs with the proposed SGD algorithms, particularly considering the previous point.

---

> ### Author Response · Authors · 2022-08-02
> **Referee 2**
>
> Below we address each question and minor questions in a point by point fashion.
>
> Weakness
>
> 1- Thank you for bringing this up. In order to consider the most general setting, we have written the existence of the kernel \Phi as an assumption (numbered 4.3) and proved that this assumption is verified in two very general and important cases (in Lemma 4.4 and Lemma 4.6). The first one deals with RKHS for H and general linear operator A. The kernel is identified explicitly in Remark 4.5 in this case. The second result deals with L^2 spaces for H and general integral operators (a very large class of operators most studied in the IP literature). For the second point (about the loss function), we thank the referee for pointing this out! Indeed, it is not necessary to consider the squared loss function. We have updated the paper accordingly (Corollary 4.4,, which is 4.8, and the proof of Theorem 4.5 (now 4.9))
>
> 2- Connecting the ML literature to SIPs allows for the latter to leverage these state-of-the-art methods to solve the ever challenging IPs in its statistical framework. We are able to show that our general framework out performs general algorithms from SIPs and practically ties with very specific methods tailored to the FLR problem. We believe that one of the main contributions is to provide a unified treatment to incorporate ML methods in many classes of SIP problems with theoretical guarantees, rather than finding a particular method that works very well in a particular problem. Moreover, our algorithm is directly applied to the online setting also. To the best of our knowledge, such type of analysis was never performed for FLR problems and can be exploit in future works.
>
> Questions:
>
> 1 - Discussed above
>
> 2- Agreed. We changed the figures of the main text and we changed the display of the results of the numerical studies. We also added additional comparisons.
>
> 3- Reference added
>
> 4- By using one sample at a time, we mean that each iteration of the algorithm makes use of only one observation in order to update the function estimated. This is in contrast with offline methods such as PFLR or Landweber iterations, that uses all the samples in every iteration. While our algorithm can also makes use of all the samples at a time (we added this discussion in the numerical studies in the appendix), our method is also suitable for online applications, that to the best of our knowledge, were not yet considered for inverse problems in such generality.
>
> 5, 6, 7 - Fixed. Thank you! we apologize for the mistake
>
> Limitations:
>
> Please, see discussions above.

---

### Official Review · Reviewer_m9Kg · 2022-07-12

**Rating:** 8
**Confidence:** 3
**Soundness:** 4 excellent
**Presentation:** 4 excellent
**Contribution:** 3 good

**Summary:**

This paper studies a method of solving functional inverse problems. The problem set-up is (X,Y) are from a particular distribution, with Y = A[f](X) + noise, where f is some function defining the relationship between X and Y. The major example is: if X is a function on [0,1], then Y = int_0^1 f(t)X(t)dt + noise. The goal of the general problem is to find f.

The paper introduces a means of finding an appropriate f from a class of functions by stochastic gradient descent, emphasizing results for a particular context: functional linear regression. They review previous results and other techniques for solving stochastic inverse problems, define the problem rigorously, and present example problems. They then give the main algorithm, a sort of functional SGD, and identify issues with it due to functional complexity involving the kernel. To address this, they provide another algorithm which uses the iteratively improving f in the SGD algorithm to teach common ML learners, which are used to build f instead of a kernel. They provide probabilistic error bounds and show a probabilistic convergence rate of O(1/ \sqrt(n)). They then present synthetic and natural experiments, with a comparison to another method.

**Questions:**

Questions:
- On Theorem 4.5, you claim the result is independent of the dimensionality of the codomain of L_2(X) and L_2(W), the values of n and k in R^d and R^k. Your proof deals with d=1. Are you sure your results are independent of the value of d? If you are sure, perhaps you could include a comment about how all inferences are independent of the values of k and d?
- Table 4 says SGD outperforms FPLR, but figure 3 indicates FPLR outperforms SGD. Is this an error? Furthermore, I find it odd that all FPLR have the same error in table 4. Are they the same function? (Aside: is SGD = SGD-SIP?)  This makes it look like some of your experimental results are incomplete or incorrect. Furthermore, the thoroughness of the deconvolution results are much less thorough than the FLR in the appendix. Please comment.
- The order of convergence for the algorithm, O(1 / \sqrt(n)), seems not so great, and the algorithm that has this guarantee looks jagged and compares poorly to the already established method. Can you comment on the order of convergence, and how it compares to other methods?

Minor issues:
- Can you provide a table for results of B2?
- Is the equation at the bottom of page 3 a component-wise integral, since X takes values in R^d but f takes values in R? This may trouble readers if not mentioned. If this is true, is it necessary X take values in R^d?
- On (138). If x is a specific realization of X, then I believe your conditioning is backwards: you want \phi(x) = E(\partial*l(Y,A[f](X))| X = x). Or do you write it the way you do because you use eq (1), with Y still a random variable of the noise, although is X specified? If this is the case, are you taking the expectation only over the noise, \epsilon? On the other hand, if x is set and Y takes one value for a specified x, what distribution is the expectation over?
- Dimensionally not accounted for in (117), (141), etc. EG: A(g)[X] in R^d, so (A[g](X))^2 not defined if d>1). Also, \partial l(...)*A[g](X) is R^d x R^d, should you should use transpose or standard inner product?
- Posed assuming the 1-d case, dropping the inner-product brackets (144,147,...). Not sure if this is notationally standard.
- (147) A*[g](w) is in R^k, but E(\Phi((X;w)g(X)) is in R^d. Even if d = 1, dimensions do not match. Is \Phi in normally in R^{k x d}?

Typos/minor issues:
- Sentence in (64-66) is confusing, hard to understand.
- "a prespecified basis functions" (74-75)
- "to control directly to tackle (3) directly" (103-104)
- "Y is giving by" (bottom of page 3)
- "also in deconvolutional problems" (112)
- indeed have a solution (128)
- does not dependent (136)
- a coarser grid where and each (214-215)
- Broken references in (380)
- Figure 3 says FPCR.

**Limitations:**

- This is my first time being exposed to the topic of the statistical inverse problem, so I am unfamiliar with some of the relevant literature.
They apply the method to standard quadratic loss, and only to FLR and deconvolution. I would like to see an exposition, even if brief, supporting the potential breadth of application. Even better, applications to more problems. This may due to my lack of familiarity with the subject.
- However, a low rate of convergence for the excess risk, and some choppy graphs for algorithm 1 seem a liability, To be fair, this is made up for with strong and interesting theoretical results that could be built of of, and a strong showing for the ML-assisted algorithm.
- Is it possible the ML-assist may not give much of a boost, or may do harm, if the kernel-sampling in not well suited to the ML-learners?

**Strengths And Weaknesses:**

Strengths:
-great organization
-clear examples
-good mathematical exposition
-solid theoretical results
-clear writing
-interesting and good experimental results
-overall interesting problem

weaknesses:
- experimental results seem inconsistently or incompletely reported.
- a few vagaries in mathematics (see questions)

---

> ### Author Response · Authors · 2022-08-02
> **Referee 1**
>
> Below we address each question and minor questions in a point by point fashion.
>
> Questions:
>
> 1-We have slightly modified the proof of the theorem to deal with any d and k. We thank the referee for this comment, we believe the proof now reades much clearer.
>
> 2- Thank you very much for catching the error in the table. To make exposition more clear, we changed the display to Error Bars across different methods. Please, see now the updated Appendix for the numerical studies. Indeed FPLR has similar performance than ML-SGD. It is actually better in our first simulated study. However, we highlight that FPLR is tailored for FLR problems, while our method is flexible enough to tackle any Linear IP problem under a statistical framework. Also, if one is focused only in the ERM problem (as PFLR is) one can make use of all the samples in every iteration of the algorithm, which increase empirical performance. We included this studied in the appendix also. We believe that one of the main contributions is to provide a unified treatment to incorporate ML methods in many classes of SIP problems with theoretical guarantees, rather than finding a particular method that works very well in a particular problem. Moreover, our algorithm is directly applied to the online setting also. To the best of our knowledge, such type of analysis was never performed for FLR problems and can be exploit in future works.
>
> 3-Unfortunately this rate is unimprovable without further assumptions for the IP. One assumption could be strong convexity of the loss function. To the best of our knowledge, this would be violated for most of the IPs, or imply very strong assumptions over the class of functions $F$, which would make the IP ''easy'' enough to be treated as a standard regression problem.
>
> Minor Questions:
>
> 1- Added in the appendix.
> 2-We made the dimensions more clearly stated throughout the paper and we updated notation/proofs to use them explicitly.
> 3-Indeed, when we are conditioning on X = x, the expectation is with respect to the noise, only.
> 4-We made the dimensions more clearly stated throughout the paper. Although some equations gets a little heavier, we believe that the operations are now more transparent.
> 5-We agree that the notation was strange for the d=1 case. We have now considered the general case through the paper.
> 6- Thanks for pointing this. Now all the dimensions are properly denoted and accounted for.
>
> Typos and minor issues:
>
> All of them were accounted for. Thank you!
>
> Limitations:
>
> 1- We addressed the issue of the L2 loss also with examples in classification problems. We also adjusted some of the Lemmas to account to a more general class of loss functions. We added a discussion about the flexibility of our approach. Please, see now remark 4.7. Our approach immediately applies to linear ODEs, some classes of PDEs and even problems in econometrics such as Non Parametric IV estimators.
>
> 2- Unfortunately, without strong convexity of the loss function with respect to f, the rate is unimprovable. While strong convexity holds in many standard regression problems, we don't believe to be the case for many linear IPs and we prefer to avoid further assumptions.
>
> 3- It is possible that the use of the ML procedure may hurt the procedure. Although we haven't see this happen in any numerical study, a function class that is not rich enough may hurt the Stochastic Gradients constructed for the IP and introduce a non-vanishing bias that might be worse than the common instability observed in standard approaches or even in the SGD-SIP that we propose.

---

### Author Response · Authors · 2022-08-02
**General Comments**

 We are extremely grateful for the positive assessment and highly detailed and constructive feedback on our paper. In this rebuttal, we have addressed the comments of the review team. This has led to multiple improvements including: better exposition of the numerical studies, new applications, more general theoretical results and improved discussions about the framework we propose. Below we address each question and minor questions in a point by point fashion.

---

### Meta-Review · Area_Chair_4r6a · 2022-08-23

**Recommendation:** Accept
**Confidence:** Certain

**Metareview:**

All reviewers recommend accepting the paper. Congratulations!

But in your camera-ready version, please revise the paper to better emphasize the relevance of this line work to a general machine learning audience. For example, during the discussion one reviewer wrote the following:

"The authors have not thoroughly convinced me of the relevance to a ML audience, as I will explain. To me, the relevance is very likely, as they apply SGD to a regression problem, and then ML learners on top of that, which is very much in the vein of ML. My confidence is limited, however, since I am not familiar with functional regression.

The paper could better appeal to an ML audience by mentioning ML applications of their method (whether it be functional linear regression or otherwise) early in their paper and in an appealing way. Section 5.2 (Real Data Application) does a good job of this, but giving motivating use cases could help pique reader interest/help them understand application earlier."

The paper will have greater influence if the final version can convince readers of its relevance to ML!

**Award:**

No

---

### Decision · Program_Chairs · 2022-09-14

Accept